# Mutational Signatures in Radiation-Induced Cancer: A Review of Experimental Animal and Human Studies

**DOI:** 10.3390/biology14091142

**Published:** 2025-08-29

**Authors:** Kazuhiro Daino, Chizuru Tsuruoka, Atsuko Ishikawa, Shizuko Kakinuma, Tatsuhiko Imaoka

**Affiliations:** 1Department of Radiation Effects Research, Institute for Radiological Science, National Institutes for Quantum Science and Technology (QST), Chiba 263-8555, Japan; tsuruoka.chizuru@qst.go.jp (C.T.); ishikawa.atsuko@qst.go.jp (A.I.); kakinuma.shizuko@qst.go.jp (S.K.); imaoka.tatsuhiko@qst.go.jp (T.I.); 2Department of Radiobiology, Institute for Environmental Sciences (IES), Aomori 039-3212, Japan

**Keywords:** biomarker, ionizing radiation, cancer risk, interstitial chromosomal deletion, chromosome rearrangement, mutational signature

## Abstract

Ionizing radiation can lead to breaks in the double-stranded structure of DNA. Although cells have mechanisms that can repair such breaks, misrejoining events can occur that lead to mutations in an individual’s genome. When mutations affect certain genes, such as oncogenes and tumor-suppressor genes, cancer can result. Although ionizing radiation is a known cause of cancer, it has been difficult to determine how much of an excess cancer risk results from low-dose radiation exposure. Radiation signatures that distinguish radiation-induced cancers from spontaneous cancers are thus useful for assessing cancer risks after exposure to low-dose radiation. Recent advances in high-throughput sequencing technologies have enabled the comprehensive profiling of mutations in individual cancer cells. Analyses of radiation-associated cancer genomes have identified an increase in small deletions and chromosome rearrangements, including large deletions, inversions, and translocations, which are likely to be formed by the misrejoining of DNA double-strand breaks. Among these, chromosome rearrangements leading to cancer-related mutational events are potentially useful as biomarkers and as therapeutic targets for treating radiation-induced cancer. This review summarizes research on radiation-induced mutational signatures found in animal and human cancers, consolidates current knowledge, and discusses the future perspectives in this research field.

## 1. Introduction

Epidemiological studies on Japanese atomic-bomb survivors, patients who have undergone radiotherapy, and workers exposed occupationally have established that ionizing radiation is a risk factor for cancer [1,2]. Although ionizing radiation is considered a non-threshold carcinogen, the cancer risk attributed to low-dose radiation is much smaller than the risk from lifestyle factors such as diet and smoking, making it difficult to accurately assess the risk by epidemiological methods, which are prone to bias problems. In fact, the results of epidemiological studies that have investigated cancer risk after exposure to low doses of ionizing radiation have not always been consistent [3]. Therefore, there is a need to identify molecular signatures that distinguish radiation-induced cancers from spontaneously developed cancers [4,5,6]. For assessing the risk of cancer after exposure to low-dose radiation, such radiation signatures must be indicative of the key event(s) in the development of radiation-induced cancer. Integrating biological data obtained using radiation signatures with epidemiology and cancer risk analysis and utilizing the integrated data to evaluate the shape of the dose–response relationships at low doses of ionizing radiation is expected to reduce the uncertainty of low-dose risk assessments. Moreover, a mechanistic understanding of carcinogenesis induced by low-dose radiation is also important for accurate estimation of cancer risk.

Cells exposed to ionizing radiation harbor a wide spectrum of DNA lesions including base damage, DNA and protein crosslinks, and DNA single-strand breaks (SSBs) and double-strand breaks (DSBs) [7]. Among the numerous forms of DNA lesions, DSBs are the most severe lesions induced by ionizing radiation, and unrejoined or misrejoined DSBs lead to cell death and mutations. A subset of mutations contributes to tumor progression (referred to as “driver” mutations), whereas the majority of mutations are effectively neutral (referred to as “passenger” mutations). Driver mutations that are directly linked to radiation-related carcinogenesis can serve as biomarkers for radiation-induced cancer and perhaps as therapeutic targets as well.

A recent analysis of large-scale cancer genome data has revealed that somatic mutations, a majority of which are passenger mutations, result in a characteristic pattern of mutations in the cancer genome, termed a mutational signature [8]. The COSMIC (Catalogue Of Somatic Mutations In Cancer) database provides sets of mutation-type-specific signatures, including single-base substitutions (SBSs), doublet-base substitutions (DBSs), small insertions and deletions (IDs), and copy number alterations (CNs) [9]. Mutational signatures in cancer genomes can give insights into past exposures to different carcinogens, as well as to the type of associated DNA damage and repair processes involved, implying the etiologies of cancer. Thus, mutational signatures can be used as biomarkers for risk assessment of radiation-induced cancers and provide valuable insights into the processes underlying tumorigenesis.

The identification of unique radiation-related biomarkers in human cancers is difficult because cancers typically result from the combined effects of radiation and other physical, chemical, and biological factors. Animal experiments are, therefore, useful for studying radiation dose responses and mechanisms of cancer development because they minimize the effects of biases and potential confounding factors. This review provides an overview of the radiation-induced mutational signatures found in animal and human cancers. In particular, we focus on signatures that are useful for assessing cancer risk after exposure to ionizing radiation.

## 2. Radiation Signatures in Animal Models of Cancer

### 2.1. Interstitial Chromosomal Deletions

Chromosomal deletion is a hallmark event directly induced by ionizing-radiation exposure. In particular, chromosomal deletions associated with specific genetic events such as the gain of oncogene activity or the loss of tumor-suppressor genes lead to the generation of cancer cells as driver mutations. Such molecular events directly link to the mechanisms of cancer formation and thus are potentially useful as a biomarker(s) for radiation-induced cancers. Studies in several rodent models of cancer have shown that interstitial chromosomal deletions of specific tumor-suppressor gene loci are observed in cancers from irradiated animals (Table 1).

#### 2.1.1. Interstitial Chromosomal Deletions in Tumors of Wild-Type Animals

An early study using a mouse model of radiation-related acute myeloid leukemia (rAML) showed the interstitial deletion of chromosome 2, including the *PU.1/Spi1* gene locus, as a characteristic genomic aberration in rAML [10]. Of note, the interstitial deletion of chromosome 2 was detected in bone marrow cells from all analyzed mice at 24 h after irradiation with 3 Gy of X-rays, suggesting that this event is the initiating molecular event that may lead to rAML [22]. In addition, a high frequency of the interstitial loss-of-heterozygosity pattern at the tumor-suppressor *Ikzf1*/*Ikaros* locus on chromosome 11 was reported to occur in murine thymic lymphomas induced by X-rays (1.6 Gy weekly exposure for 4 weeks, with a total dose of 6.4 Gy), as compared with spontaneously developed or alkylating agent–induced lymphomas [11]. Subsequent studies also identified interstitial deletions of chromosomes 4 and 19, including the *Cdkn2a* and *Pten* loci, respectively, in radiation-induced thymic lymphomas [12,13]. In contrast, interstitial chromosomal deletions were rarely observed in thymic lymphomas from X-irradiated (2 Gy) mice that were deficient for the DNA mismatch repair gene *Mlh1* [23]. This is probably attributable to the major contribution to lymphomagenesis of a high burden of single-nucleotide variants (SNVs) and insertions/deletions (InDels) that result from mismatch repair deficiency. These observations suggest that, except in cases such as the mismatch repair abnormalities, interstitial chromosome deletions at specific tumor-suppressor gene loci are a characteristic genomic feature of radiation-induced tumors.

Consistent with these earlier reports on rAML and thymic lymphoma, more recent studies have also identified interstitial chromosomal deletions of specific tumor-suppressor gene loci in tumors that developed in wild-type animals after radiation exposure. In B6C3F1 mice, interstitial deletion of chromosome 4, including the tumor-suppressor *Pax5* locus, was specifically found in early-onset precursor B-cell lymphoma induced in mice after 4 Gy of gamma-ray exposure [14]. In addition, interstitial deletion of chromosome 5, including the tumor-suppressor *Cdkn2a* locus, was reported in mammary carcinomas from Sprague-Dawley rats irradiated with 2 or 4 Gy of gamma-rays or 1 Gy of fast neutrons [15,16]. These findings highlight the generalizability of interstitial chromosomal deletions of specific tumor-suppressor gene loci in radiation-induced cancers. Further investigation is needed to assess whether these findings are relevant to human cancer.

#### 2.1.2. Interstitial Chromosomal Deletions in Tumors from Heterozygous Mutant Animals

Several heterozygous mutant rodent models have been used for cancer studies because of their susceptibility to radiation carcinogenesis. The *Ptch1* heterozygous (*Ptch1*^+/−^) mouse is an established model of human Gorlin syndrome that is sensitive to the carcinogenic effect of ionizing radiation [24]. In this mouse model, the interstitial deletion of chromosome 13, which harbors the *Ptch1* locus, occurs in medulloblastomas (MBs) induced by X-ray exposure [17,18]. Analysis of the loss-of-heterozygosity patterns in MBs from these mice indicated that spontaneous loss of the wild-type *Ptch1* allele in MBs from non-irradiated mice is mediated by mitotic recombination or non-disjunction (referred to as a spontaneous tumor), whereas radiation-associated loss is mediated by interstitial deletion (referred to as a radiation-induced tumor) (Figure 1A). In addition, this study provided evidence for a radiation dose–dependent increase in MBs harboring the interstitial deletion in mice exposed to 0.05–3 Gy of X-rays (Figure 1B). These data indicated that the interstitial chromosomal deletion of the *Ptch1* locus is useful as a radiation signature for evaluating the carcinogenic effects of low-dose radiation exposure.

The generalizability of interstitial chromosomal deletions of specific tumor-suppressor gene loci in radiation-induced cancers was also demonstrated in the following animal model tumors. The Eker rat, a model of human tuberous sclerosis complex, which carries a heterozygous germline mutation in the tumor-suppressor gene *Tsc2*, is susceptible to radiation-induced renal carcinogenesis [19]. Interstitial deletion of chromosome 10, including the *Tsc2* locus, occurs in kidney tumors from Eker rats irradiated with 2 Gy of gamma-rays [20]. The *Apc*^Min/+^ mouse, with a heterozygous germline mutation of *Apc*, is one of the most widely used models for human familial adenomatous polyposis and is predisposed to radiation-induced intestinal tumorigenesis [25,26]. In gamma-irradiated (2 Gy) *Apc*^Min/+^ mice, interstitial deletion of chromosome 18, including the tumor-suppressor gene *Apc*, was observed in intestinal tumors [21]. Notably, the interstitial chromosomal deletion of the *Apc* locus was detected only in cells that overexpressed β-catenin, indicating the importance of the analysis with respect to intratumor heterogeneity for the identification of radiation signatures.

#### 2.1.3. Cancer-Risk Assessment Using Interstitial Deletion

Interstitial chromosomal deletions have been used for cancer-risk assessment for low-dose radiation exposure using the *Ptch1*^+/−^ mouse model. MBs that developed in *Ptch1*^+/−^ mice that were not irradiated or were irradiated with a low dose (or at a low dose rate) were classified into spontaneous and radiation-induced tumors by detecting the interstitial deletion at the *Ptch1* locus as a radiation-specific signature [27]. This study detected a reduced incidence of radiation-induced MBs after low-dose gamma-ray exposure (100 mGy) as compared with high-dose exposure (500 mGy) (Figure 2). In addition, there was a reduced cancer risk from protracted low dose rates of exposure (5.4 mGy/h or 1.1 mGy/h) as compared with the same total dose of acute irradiation (100 mGy or 500 mGy, respectively). Thus, the *Ptch1*^+/−^ mouse has provided a useful model for directly assessing the cancer risk from low-dose and low-dose-rate radiation exposure by using a genomic signature of radiogenic cancer. Further evaluation of various types of tumors and radiation exposure is needed in the future.

### 2.2. Genome-Wide Mutational Signatures

Genome-wide profiling by high-throughput sequencing has been conducted using several tumors that developed in animals exposed to ionizing radiation (Table 2). The whole-exome sequencing (WES) analysis of rat mammary tumors found no significant difference in the number of SNVs or InDels per tumor among tumors that developed in gamma ray–irradiated (4 Gy), fast neutron–irradiated (1 Gy), or non-irradiated animals [16]. Similarly, an analysis of the COSMIC mutational signatures did not reveal significant differences among those tumors. Consistent results were reported in a WES analysis of thymic lymphomas from *Trp53*-deficient mice that received fractionated radiation (1.8-Gy weekly fractions for 4 weeks, total dose of 7.2 Gy) [28]. Interestingly, radiation-associated mutational signatures enriched in C-to-T or T-to-G substitutions were reported in WES studies of several types of tumors, including soft tissue sarcomas, squamous cell carcinomas, mammary carcinomas, and hematopoietic malignancies, from wild-type or *Nf1^+/−^* mice that had been subjected to fractionated irradiation (3 Gy daily fractions for 5 days a week, total dose of 30 Gy) [29,30]. These data suggest that radiation-specific mutation patterns may be difficult to identify from the limited data obtained for exon regions and that such patterns are observed primarily for certain genotypes or exposure conditions, such as fractionated irradiation, which results in high cumulative doses.

Whole-genome sequencing (WGS) analysis can identify variants in non-coding regions, genomic copy-number changes, and structural variants (SVs) in cancer cells, none of which are readily detected by WES analysis. A WGS study of mammary tumors that developed in *Trp53*-deficient mice after exposure to 0.5 Gy of high linear energy transfer (LET) radiation (^56^Fe ions) or low-LET radiation (gamma-rays) showed that ^56^Fe ion-induced tumors are associated with the induction of focal SVs, leading to genomic instability and *Met* amplification [31]. This study also showed that the COSMIC mutational signature SBS18, which is associated with oxidative DNA damage and characterized by enrichment of C-to-A transversions, is prevalent in tumors from gamma-ray–irradiated animals, suggesting the involvement of reactive oxygen species –mediated mutagenesis by gamma-ray exposure. Interestingly, these findings suggest the existence of radiation type–dependent mutational signatures in radiation-induced cancers.

Although there is cumulative evidence for genome-wide mutational signatures of radiation-related animal tumors, it is still difficult to interpret those data owing to differences in experimental design, the rodent models used, and radiation type. In addition, a comparison of genomic profiles between tumors that develop after exposure to ionizing radiation and tumors induced by other causes is also needed to identify radiation-specific events.

## 3. Radiation Signatures in Human Cancers

### 3.1. Genomic Alterations

A loss or gain of genomic DNA is frequently observed in cancer cells, and these alterations can affect the function of tumor-suppressor genes or oncogenes, respectively. Several studies have reported characteristic genomic copy-number alterations in radiation-associated cancers that develop in humans (Table 3).

#### 3.1.1. Genomic Alterations in Hematopoietic Neoplasms

An early study using fluorescence in situ hybridization (FISH) analysis indicated a higher incidence of monosomy 7 and of deletions at chromosomal region 20q13.2 in AMLs among atomic-bomb survivors as compared with AMLs from unexposed individuals [32]. In addition, point mutations in *AML1/RUNX1* were observed at a high frequency in myelodysplastic syndrome (MDS)/AML cases in radiation-exposed residents near the Semipalatinsk Nuclear Test Site as compared with MDS/AMLs from unexposed individuals [33]. A study of chronic lymphocytic leukemia (CLL) in Chornobyl clean-up workers (Table 3) revealed more frequent mutations in the telomere-maintenance pathway genes *POT1* and *ATM* as well as significantly longer telomere length in radiation-exposed CLL cases as compared with CLL in unexposed individuals [34]. In addition, genomic analysis of myeloproliferative neoplasms (MPNs) in a total of 90 patients exposed during the Chornobyl accident (Table 3) found different mutation frequencies in *JAK2*, *CALR*, *ATM*, *EZH2*, and *SUZ12* in MPNs from exposed patients as compared with those from unexposed patients [35]. These observations reveal frequent chromosomal copy-number changes and mutations in several genes in radiation-related hematologic malignancies, suggesting their potential as biomarkers. However, it will be necessary to investigate whether these alterations can be caused directly by radiation.

#### 3.1.2. Genomic Alterations in Thyroid Cancer

Previous genomic DNA copy-number studies of papillary thyroid cancers (PTCs) from young patients exposed to Chornobyl fallout reported DNA copy-number gains in the chromosome region 7q11.22–11.23 [36]; further, overexpression of *CLIP2*, which is located in this region, was apparent at both the mRNA and protein levels in PTCs from radiation-exposed individuals as compared with unexposed cases [37,38]. Results from these analyses suggest that *CLIP2* overexpression may serve as a potential biomarker for radiation-induced PTCs, although it is unclear whether this alteration is directly induced by radiation exposure.

#### 3.1.3. Genomic Alterations in Breast Cancer

Another cytogenetic study found frequent amplification of the oncogenes *c-MYC* and *HER2* and association of these two amplifications with higher histological grade in breast cancers among atomic-bomb survivors in Japan [39]. Subsequent DNA copy-number analysis also revealed genomic instability in breast cancers isolated from atomic-bomb survivors [40]. A study of breast cancers from 68 patients who were exposed to ionizing radiation as Chornobyl clean-up workers and evacuees (Table 3) identified a genomic copy-number alteration (CNA) signature associated with radiation exposure [41], which included the copy number gain at 7q11.22–11.23 that was observed in PTCs. There was, however, no dose–response relationship relative to the occurrence of the CNA signature in this study, which might be attributable to several possibilities including the limited number of patients for whom dose estimation data were available and human-factor uncertainty. In contrast to the study of atomic-bomb survivors, this analysis of Chornobyl workers and evacuees failed to detect an association between *c-MYC* and *HER2* oncogene amplification and high histological grade. This could be attributable to the more heterogenous radiation exposure and younger age at the time of diagnosis among individuals who developed post-Chornobyl breast cancers as compared with the Japanese atomic-bomb survivors. In addition, a large-scale study of 77 radiotherapy-induced sarcomas reported a higher frequency of *c-MYC* amplification and losses of *CDKN2A* and *CDKN2B* in radiotherapy-induced sarcomas than in sporadic sarcomas [42]. These observations indicate that radiation-associated breast cancers have characteristic chromosomal copy-number changes and a high frequency of *c-MYC* and *HER2* oncogene amplification. However, further investigation is necessary to verify whether these alterations are directly related to radiation exposure.

Although the characteristic genomic alterations found in several radiation-related human cancers differ from those in spontaneous cancers, these studies were not able to identify specific driver-gene mutations or other genetic alterations that could be considered biomarkers for radiation-induced tumors.

### 3.2. Genome-Wide Mutational Signatures

Recent cancer genome sequencing studies revealed mutational signatures in radiation-associated cancers that developed in patients who received radiotherapy or in residents and workers exposed to ionizing radiation after nuclear accidents (Table 4). A WGS study of radiation-associated cancers reported a high frequency of small deletions and balanced chromosomal inversions in radiotherapy-induced second cancers, including sarcoma and breast cancer, as compared with radiation-naive cancers [43]. Consistent with these findings, frequent SVs and the COSMIC mutational signature ID8—characterized by small deletions and proposed to arise from classical non-homologous end joining (c-NHEJ) of radiation-induced DSBs [44]—were found in a recent WGS study of radiotherapy-induced sarcomas [45]. These observations suggest an association between c-NHEJ–mediated DSB repair and the tumorigenesis of radiation-induced second cancers after radiotherapy.

A unique resource for studying the genomic characteristics of human cancers in individuals with chronic exposure to high-LET radiation has been provided by an analysis of workers who were occupationally exposed to alpha radiation emitted by plutonium at the Mayak Production Association in the former Soviet Union. WES analysis of liver tumors, including angiosarcoma, cholangiocarcinoma, and hepatocellular carcinoma, from the Mayak employees indicated an excess of small deletions and clustered mutations (one to three SNVs within 5 base pairs of one another) as well as driver gene mutations involved in actin cytoskeletal signaling and DSB repair [46]. These data suggest the existence of characteristic genomic alterations in liver tumors induced by high-LET alpha-radiation, although these changes must be further validated because of the small sample size in the study.

A recent large-scale WGS study reported genomic, epigenomic, and transcriptomic alterations in a total of 359 PTCs (Table 4) related to the Chornobyl accident [47]. Although no relationship between *CLIP2* expression and radiation dose was observed, this study did show a radiation dose–dependent increase in small deletions and simple/balanced SVs, which were likely caused by NHEJ repair. Furthermore, radiation dose–dependent increases in the COSMIC mutational signature ID8 and fusion driver mutations—mainly in genes involved in the mitogen-activated protein kinase signaling pathway—were also found in the PTCs. This large-scale comprehensive study identified radiation-associated mutational signatures in PTCs by showing their dose-dependent increase. Although no radiation-specific alterations were apparent in this research, further investigations are needed with other types of tumors.

### 3.3. Clinical Implications of Mutational Signatures

Mutational signatures in radiation-induced cancers are paramount for accurate diagnosis, prognosis, and treatment selection. The aforementioned small deletions and SVs observed in radiotherapy-associated second cancers could be used as biomarkers for diagnosis and prognosis of cancers in clinical practice. Interestingly, a study of radiation-induced gliomas reported recurrent *PDGFRA* amplification and loss of *CDKN2A/B* in the absence of histone *H3* or *IDH1/2* mutations as diagnostic markers and potential targets for treatment [48]. In addition, a recent large-scale WGS study using a total of 156 recurrent gliomas arising in patients after radiotherapy found small deletion burden and homozygous deletion of *CDKN2A* as biomarkers of short survival and recurrence after radiotherapy, respectively [49]. The small deletion signature, which is characteristic of NHEJ-mediated DSB repair, also suggests that inhibition of NHEJ could sensitize tumors to radiotherapy.

On the other hand, it should be noted that various therapeutic approaches exist to reduce the risk of recurrent and secondary cancers after radiotherapy by enhancing cancer-cell killing and minimizing DNA damage in normal cells around the tumor. These include protracted radiotherapy and methods such as the use of radioprotectors, fractionation and dose-rate optimization, ultra-high-dose-rate FLASH radiotherapy (FLASH-RT), advanced modalities like intensity-modulated radiotherapy, stereotactic body radiotherapy, tomotherapy, proton and heavy-ion therapies, and biological modulation of DNA repair and immunity via gene- and RNA-based strategies [50].

## 4. Discussion and Future Perspectives

Radiation signatures that distinguish radiation-induced cancers from spontaneous cancers are useful for not only assessing cancer risk after low-dose radiation exposure but also understanding the molecular mechanisms of radiation carcinogenesis. To date, our knowledge of mutational signatures in radiation-induced cancer is steadily increasing. The data from human cancers show that small deletions and chromosomal rearrangements, including large deletions, inversions, and translocations, are radiation-associated genomic alterations that are possibly involved in the generation of driver mutations. However, driver gene mutations unique to radiation exposure have not been found in these cancers. In addition, the genomic alterations of radiation-associated cancer in humans have been reported for only a limited number of cancer types. Therefore, further investigation of radiation-specific driver events in different types of cancers will be necessary. In this regard, studies on organs relevant to radiological protection (e.g., bone marrow, digestive tract, lung, and mammary gland) are particularly important.

Animal models of cancer are invaluable experimental systems in which to study different cancer types as well as the molecular mechanisms underlying radiation-induced cancer. Among other advantages is the ability to control radiation doses precisely in such systems. Interstitial chromosomal deletions at specific tumor-suppressor gene loci have commonly been found in several types of tumors that develop in animals after radiation exposure, suggesting the generalizability of the interstitial chromosomal deletion in radiation-induced cancers. In addition, interstitial chromosomal deletions affecting cancer-driver genes are directly linked to the mechanisms of cancer development and therefore are potentially useful as biomarkers for cancer-risk assessment after low-dose radiation exposure. We note, however, that although recent genome-wide sequencing studies have described mutational signatures in radiation-related tumors in several animal models, it is still difficult to interpret those data due to differences in the experimental design, the mouse models used, and types of radiation. A comparison of relevant data from various types of tumors obtained from laboratory animals will clarify the generalizability or specificity of radiation signatures and will support human studies.

For assessment of low-dose radiation risk, it is important to compare the observed mutation spectrum at low doses with that at high doses. However, only a few studies have investigated the mutation spectra generated at low doses. A study using a sensitive system to detect somatic mutations at the *HPRT1* locus in rodent cells reported that the mutation spectrum induced by low-dose X-rays was similar to that of spontaneous mutations, but a significant change in the type of mutations was observed at doses exceeding 0.2 Gy [51]. A genomic study of mammary tumors induced by low-dose (0.5 Gy) of Fe-ion or gamma ray irradiation in *Trp53*-deficient mice identified SNV signatures associated with homologous recombination DNA repair defects and the abundance of reactive oxygen species (ROS), respectively, as well as InDel signatures [31]. The study of CLLs in Chornobyl cleanup workers, including those exposed to low-dose radiation, reported a failure to identify SNV signature that observed at high-dose radiation (total dose of 30 Gy)-induced malignant tumors from *Nf1*^+/−^ mice [34]. On the other hand, a study of PTCs following the Chornobyl nuclear accident demonstrated a dose-dependent increase in radiation-associated mutational signatures, but no differences were identified between low-dose and high-dose exposure [47]. Therefore, in addition to studying the mutation spectra observed at low doses, detailed studies are necessary that consider factors such as the differences in protracted exposure to low-dose radiation versus acute irradiation.

Radiation signatures for cancer-risk assessment are also useful for the evaluation of reference values used in radiation protection. In this regard, in the *Ptch1^+/^*^−^ mouse model, interstitial chromosomal deletions have been used as a biomarker of radiogenic cancer for the accurate calculation of relative biological effectiveness (RBE) values, defined as the ratio of the photon dose to the dose of another type of ionizing radiation that is required to achieve the same biological effect [52,53]. These studies have provided accurate calculations of RBE values for fast neutrons, such as the secondary neutrons generated in proton, heavy-particle, and intensity-modulated radiation therapies [54,55,56], and low-LET carbon ions with respect to the induction of MB, which is useful for assessing the risk of second cancer from radiation exposure of normal tissues in patients treated with modern radiotherapy.

Radiotherapy is a powerful tool for cancer treatment, but it can also damage the DNA of normal cells surrounding the tumor, thus contributing to the development of second cancers. In contrast, hypoxic conditions within the tumor microenvironment are known to suppress the generation of ROS, thereby attenuating the cytotoxic effects of radiation [57]. Unfortunately, this enables the survival and proliferation of residual cancer cells, leading to tumor recurrence. Because primary and secondary cancers occur via different mechanisms, they are expected to have unique mutational signatures. Therefore, the differences in these mutational signatures may provide a basis for elucidating the mechanisms of cancer development after radiotherapy and inform treatment strategies.

Although studies of radiation-induced cancers have revealed characteristic mutational signatures, the mutation spectra directly induced by ionizing radiation—particularly in normal cells—have not been documented comprehensively. Interestingly, recent WGS studies of single-cell-derived colonies have revealed radiation-associated mutations, including short deletions and simple and complex SVs, in normal mammalian cells [58,59]. These data will help clarify those mutations directly induced by ionizing-radiation exposure and inform discussions of their association with the mutational signatures observed in cancer. The rapid development of new technologies has allowed comprehensive and highly sensitive analyses of molecular changes in cancer cells. In this regard, single-cell genomics and multi-omics technologies could lead to the identification of radiation-specific signatures and related molecular networks. Applying these technologies to various cancer types will identify new or novel classes of biomarkers as well as facilitate our understanding of carcinogenic mechanisms after exposure to low-dose radiation.

## 5. Conclusions

Recent advances in genomic technologies have significantly enhanced our understanding of radiation-induced cancers. The identification of radiation-associated mutational signatures—such as small deletions and chromosomal rearrangements—offers a promising avenue for distinguishing radiation-induced cancers from spontaneous cancers. Although these alterations may play a role in generating driver mutations, no radiation-specific driver-gene mutations have yet been identified in human cancers. Continued research across diverse cancer types relevant to radiological protection will be essential for improving cancer-risk assessments, diagnoses, and treatment strategies and understanding the molecular mechanisms of radiation carcinogenesis.

## Figures and Tables

**Figure 1 biology-14-01142-f001:**
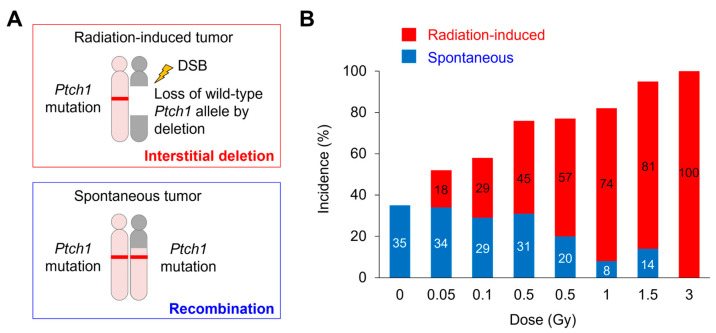
Radiation dose–dependent increase in interstitial chromosomal deletion at the *Ptch1* locus in MBs from *Ptch1*^+/−^ mice. (**A**) Radiation-associated loss of the wild-type *Ptch1* allele in MB is mediated by interstitial deletion (radiation-induced), whereas spontaneous loss is mediated by mitotic recombination or non-disjunction (spontaneous). (**B**) Dose-effect relationships of the incidence of spontaneous (blue) and radiation-induced (red) MBs in mice. DSB, double-strand break. Modified based on the data in Ishida et al. [18].

**Figure 2 biology-14-01142-f002:**
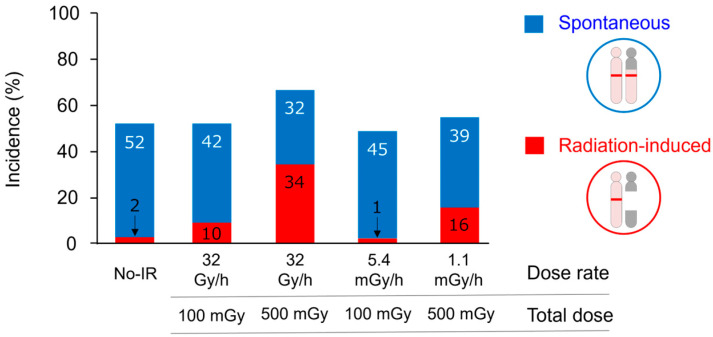
Sensitive detection of radiation-induced MBs after low-dose or low-dose-rate radiation exposures in *Ptch1^+/−^* mice using a radiation-specific molecular signature. Data represent the incidences of spontaneous (blue) and radiation-induced (red) tumors in *Ptch1^+/−^* mice that were not irradiated (No-IR) or were irradiated with the indicated dose and dose rate of gamma rays. Modified based on the data in Tsuruoka et al. [27]. Numbers in bar graphs indicate the incidence values in percentage.

**Table 1 biology-14-01142-t001:** Interstitial chromosomal deletions of specific tumor-suppressor gene loci found in animal tumors developed after ionizing-radiation exposure.

Tumor Type	Animal Model	Genotype	Method	Dose	Deletion Target Gene (chr)	Reference(s)
AML	(CBA/H × C57BL/Lia) F1 mouse	Wild type	LOH	3 Gy	*PU.1/Spi1* (2)	[10]
Thymic lymphoma	(C57BL/6 × C3H/He) F1 mouse	Wild type	LOH and array-CGH	6.4 Gy (1.6 Gy × 4 weekly fractionated X-ray)	*Cdkn2a* (4)*, Ikzf1* (11)*, Pten* (19)	[11,12,13]
B-cell lymphoma	(C57BL/6 × C3H/He) F1 mouse	Wild type	LOH, array-CGH, and WES	4 Gy	*Pax5* (4)	[14]
Mammary carcinoma	Sprague-Dawley rat	Wild type	Array-CGH and WES	2 or 4 Gy, 1 Gy (fast neutron)	*Cdkn2a* (5)	[15,16]
Medulloblastoma	*Ptch1*^+/−^ mouse	Heterozygous loss-of-function mutation	LOH and array-CGH	0.05 Gy to 3 Gy	*Ptch1* (13)	[17,18]
Renal carcinoma	*Tsc2*^Eker/+^ rat	Heterozygous loss-of-function mutation	LOH and array-CGH	2 Gy	*Tsc2* (10)	[19,20]
Intestinal carcinoma	*Apc*^Min/+^ mouse	Heterozygous loss-of-function mutation	LOH and array-CGH	2 Gy	*Apc* (18)	[21]

Abbreviation: AML, acute myeloid leukemia; Array-CGH, microarray-based comparative genomic hybridization; chr, chromosome; LOH, loss of heterozygosity; WES, whole-exome sequencing.

**Table 2 biology-14-01142-t002:** Mutational signatures found in animal tumors that developed after ionizing-radiation exposure.

Tumor Type(s)	Animal Model	Method	Dose	Characteristic Alterations	Reference(s)
Sarcoma, squamous cell carcinoma, mammary carcinoma, and hematopoietic neoplasm	*Nf1*^+/−^ and wild-type mice	WES	30 Gy (3 Gy × 5, daily fractionated X-ray)	DNA copy-number loss (*Nf1*^+/−^ mice) and mutational signatures enriched for C-to-T or T-to-G substitutions	[29,30]
Mammary carcinoma	*Trp53*^+/−^ and *Trp53ΔP* mice	WGS	0.5 Gy (Fe-ion or gamma-ray)	Focal SVs and *Met* (chr 6) amplification (Fe-ion), SVs and COSMIC signature SBS18 associated with ROS (gamma-ray)	[31]

Abbreviations: COSMIC, Catalogue Of Somatic Mutations In Cancer; chr, chromosome; ROS, reactive oxygen species; SVs, structural variants; WGS, whole-genome sequencing; SBS, single-base substitution.

**Table 3 biology-14-01142-t003:** Genome copy-number alterations found in radiation-associated human cancers.

Tumor Type	Cohort	Method	Dose	Characteristic Alterations	Reference(s)
AML	Atomic-bomb survivors	FISH	N.A.	Monosomy 7 and chromosomal deletion at 20q13.2	[32]
MDS/AML	Residents near Semipalatinsk Nuclear Test Site	PCR-SSCP ^7^	N.A.	*AML1*/*RUNX1* (chr 21) point mutations	[33]
CLL	Chornobyl clean-up workers	Targeted deep sequencing of 538 cancer-relevant genes and off-target reads mapping	Median bone marrow dose, 40.5 mGy; range, 0.4 to 1536.2 mGy	*POT1* (chr 7) and *ATM* (chr 11) mutations and longer telomere length	[34]
MPN	Chornobyl clean-up workers and residents	WES	20 to 500 mSv (cleanup workers) and 5.9 to 31 mSv (residents)	*JAK2* (chr 9), *CALR* (chr 19), *ATM* (chr 11), *EZH2* (chr 7), and *SUZ12* (chr 17) mutations	[35]
Papillary thyroid cancer	Residents after Chornobyl accident	Array-CGH, real-time qPCR, and IHC	Mean dose, 0.15 to 1.2 Gy	DNA copy-number gain of 7q11.22-11.23 and *CLIP2* (chr 7) overexpression	[36,37,38]
Breast cancer	Atomic-bomb survivors	FISH and array-CGH	N.A.	*C-MYC* (chr 8) and *HER2* (chr 17) amplification	[39,40]
Breast cancer	Chornobyl clean-up workers and evacuees	Array-CGH	Median dose, 13.0 mGy; range, 0.06 to 582.9 mGy (clean-up workers) and median dose, 18.4 mGy; range, 5.72 to 36.6 mGy (evacuees)	CNA signature consists of chromosome regions at 7q11.22–11.23, 7q21.3, 16q24.3, 17q21.31, 20p11.23–11.21, 1p21.1, 2q35, and 6p22.2	[41]
Sarcoma	Patients who received radiotherapy	Array-CGH	N.A.	*C-MYC* (chr 8) amplification and *CDKN2A* (chr 9) and *CDKN2B* (chr 9) losses	[42]

Abbreviations: AML, acute myeloid leukemia; array-CGH, microarray-based comparative genomic hybridization; CLL, chronic lymphocytic leukemia; CNA, copy-number alteration; chr, chromosome; FISH, fluorescence in situ hybridization; MDS, myelodysplastic syndrome; MPN, myeloproliferative neoplasm; N.A., not available; PCR-SSCP, polymerase chain reaction–single-strand conformation polymorphism.

**Table 4 biology-14-01142-t004:** Mutational signatures found in radiation-associated human cancers.

Tumor Type	Cohort	Method	Dose	Characteristic Alterations	Reference
Sarcoma and breast cancer	Patients who received radiotherapy	WGS	N.A.	Small deletions and balanced chromosomal inversions	[43]
Sarcoma	Patients who received radiotherapy	WGS	45 to 54 Gy (25 to 30 fractionated), including unknown cases	Small deletions, COSMIC signature ID8 and SVs	[45]
Liver tumor	Mayak employees	WES	N.A.	Small deletions and clustered mutations	[46]
Papillary thyroid cancer	Residents after Chornobyl accident	WGS, mRNA/microRNA-seq, DNA methylation profiling	Mean dose, 250 mGy; range, 11 to 8800 mGy	Small deletions and simple/balanced SVs, COSMIC signature ID8, and fusion driver mutations	[47]
Glioma	Patients who received radiotherapy	WES, mRNA-seq, DNA methylation profiling	N.A.	*PDGFRA* (chr 4) amplification and *CDKN2A* (chr 9) and *CDKN2B* (chr 9) losses	[48]
Glioma	Patients who received radiotherapy	WGS	N.A.	Small deletions, COSMIC signature ID8, and chromosomal deletions and inversions, and *CDKN2A* (chr 9) loss	[49]

Abbreviations: COSMIC, Catalogue Of Somatic Mutations In Cancer; chr, chromosome; N.A., not available; SVs, structural variants; WES, whole-exome sequencing; WGS, whole-genome sequencing.

## Data Availability

No new data were created or analyzed in this study. Data sharing is not applicable to this article.

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
