# Peer review of "Mutational Signatures in Radiation-Induced Cancer: A Review of Experimental Animal and Human Studies"

_biology, 2025, doi:10.3390/biology14091142_

Round 1
Reviewer 1 Report
Comments and Suggestions for Authors
In this review, Daino et al. provides an overview of radiation-induced mutational signatures in animal and human cancers with the focus on signatures that are useful in assessing cancer risks after ionizing-radiation exposure. The review covers most studies in the field and contains relevant and adequate details. The reviewer has the following comments:
1. The first paragraph of the introduction could better explain why identifying molecular signatures is important for low-dose radiation risk assessment.
2. Some terms like "passenger mutations" need to be explained the first time it is mentioned to increase clarify.
3. The connection between mutational signatures and cancer risk assessment could be made more explicit.
4. Under 'radiation signatures in animal cancer', some parts provide very specific details like doses, locations etc. while others are more general. Please maintain technical detail balance throughout the text.
5. The last paragragh in section 2.1.2 seems disconnected and needs to be better integrated into the text.
6. Under section 3 (radiation signatures in human cancer)-there is a quick jump from general genomic alteration to specific example of thyroid cancer. Please provide adequate context here as to why you are looking specifically into thyroid cancer for better clarity.
7. Additionally, there needs to be some improvement in the flow as the text jumps between different cancer types and lacks clear logical transitions.
8. Overall, section 3.1 provides details of a lot of findings from different studies but there is no effective synthesis of how different genomic alterations collectively contribute to our understanding of radiation- induced cancer. Concluding statements highlighting this point should be provided.
9. The discussion section lacks clear and specific recommendations for future technologies/approaches that might be promising for identifying radiation signatures. Including this would strengthen the section.
The reviewer has the following comments:
1. There is lot of redundancy present throughout the text specifically in the introduction. Please reduce redundancy as much as possible.
2. Transitions between paragraghs are sometimes abrupt and lacking context eg. -transition between 2nd and 3rd paragragh of introduction. Better flow is expected.
3. Additionally consider breaking longer paragraghs for better readability.
4. Please correct minor grammatical errors.
Reviewer 2 Report
Comments and Suggestions for Authors. The manuscripts are well written by authors, and providing literature in table form helps the reader to get it easily.
However, there are a few areas where the author can provide depth and more clarity. Please consider the following points:
Comments
- The figure on page#2 needs to be improved and made it more clear with a change in font size. The figure does not have a title or brief details. It will help to new reader to get knowledge about radiation and types of alterations in DNA.
- I would suggest trying to expand the literature on the clinical implications side. Like how these signatures would be helpful to inform diagnosis and treatment.
- Adding more relevant references for these studies would strengthen them.
- Before resubmission, go through once to check for small grammar errors and sentences.

Reviewer 3 Report
Comments and Suggestions for Authors
This manuscript (title: Mutational Signatures in Radiation-Induced Cancer: a Review of Experimental Animal and Human Studies) reexamines existing data and summarizes research on radiation-induced mutational signatures found in animal and human cancers, consolidates current knowledge, and discusses the future perspectives.
There are some issues needed further attention.
1. In the second paragraph of Introduction authors mentioned that “Cancer is a genetic disease that arises from the accumulation of various mutations and chromosome aberrations”. Although all cancers involve genetic mutations, but using the term "genetic disease" might be misleading when applied to cancer. It's more appropriate to say that cancer is a genetic disease, but it's not always an inherited one. For instance, cancer arises from DNA damage leads to uncontrolled cell growth, and this damage can be either inherited or acquired during a person's lifetime.
2. Authors used non relevant references at several places for instance in the heading 2.1.1. “Sprague-Dawley rats irradiated with 4 Gy of gamma-rays or 1 Gy of fast neutrons [17,26]”. Only reference 26 is correct reference 17 used 2 Gy. In the heading 2.1.2. “The Ptch1 heterozygous (Ptch1+/–) mouse is an established model of human Gorlin syndrome that is sensitive to the carcinogenic effect of ionizing radiation [30,31]”. Here Reference 30 is not about the Gorlin Syndrom. Authors should check entire manuscript carefully for non-relevant references.
3. In the Figure-1B authors presents % tumor incidence data and used terms Radiation type and Spontaneous type. Radiation type should be replaced with radiation induced, since radiation type can be confused with different type of radiation like Low-LET and High-LET, for better readership.
4. The authors should conclude each paragraph with a brief summary and discussion of future research directions to enhance the manuscript's clarity and focus.
Reviewer 4 Report
Comments and Suggestions for Authors
This review comprehensively summarizes the mutational signatures of radiation-induced cancer identified in animal and human epidemiological studies. It is well-arranged but there are some points to supplement.
1. It would be beneficial to arrange reference that compares the frequency of mutation signatures observed at low doses with those observed at high doses (>1 Gy). Alternatively, reliable data on typical average radiation doses or periods related to occuring mutation is suggested. Generally known that the mutation would be higher by acute radiation exposure compared to the low-dose.
2. Radiation provides therapeutic benefits in humans because it directly induces the death of cancer cells. Therefore, the fundamental causes of secondary cancers should be discussed separately, considering genetic variations in normal cells and the limitations of a decrease in cancer cell viability, such as hypoxia.
3. If the risk of mutations exists, various methods exist to prevent and treat it. These methods include protraction radiation, and the rest can be summarized as follows: a. utilization of radioprotectors; b. fractionation and dose-rate optimization; c. FLASH radiotherapy; d. application of different modalities such as intensity-modulated radiotherapy (IMRT/VMAT/IGRT), stereotactic body radiotherapy, tomotherapy, CyberKnife, proton and heavy-ion therapy; e. Biological Modulation of DNA Repair and Immunity, such as Gene- and RNA-based strategies. It is suggested to organize these contents into a separate section.
